# Bromodomain (BrD) Family Members as Regulators of Cancer Stemness—A Comprehensive Review

**DOI:** 10.3390/ijms24020995

**Published:** 2023-01-04

**Authors:** Patrycja Czerwinska, Andrzej Adam Mackiewicz

**Affiliations:** 1Department of Cancer Immunology, Poznan University of Medical Sciences, 61-866 Poznan, Poland; 2Department of Diagnostics and Cancer Immunology, Greater Poland Cancer Centre, 61-866 Poznan, Poland

**Keywords:** bromodomain, epigenetic factor, cancer stem cells, self-renewal, pluripotency, cancer stemness, TRIM28, ATAD2, BRD4

## Abstract

Epigenetic mechanisms involving DNA methylation and chromatin modifications have emerged as critical facilitators of cancer heterogeneity, substantially affecting cancer development and progression, modulating cell phenotypes, and enhancing or inhibiting cancer cell malignant properties. Not surprisingly, considering the importance of epigenetic regulators in normal stem cell maintenance, many chromatin-related proteins are essential to maintaining the cancer stem cell (CSC)-like state. With increased tumor-initiating capacities and self-renewal potential, CSCs promote tumor growth, provide therapy resistance, spread tumors, and facilitate tumor relapse after treatment. In this review, we characterized the epigenetic mechanisms that regulate the acquisition and maintenance of cancer stemness concerning selected epigenetic factors belonging to the Bromodomain (BrD) family of proteins. An increasing number of BrD proteins reinforce cancer stemness, supporting the maintenance of the cancer stem cell population in vitro and in vivo via the utilization of distinct mechanisms. As bromodomain possesses high druggable potential, specific BrD proteins might become novel therapeutic targets in cancers exhibiting de-differentiated tumor characteristics.

## 1. Introduction

Tumors are complex ecosystems in which distinct cell populations with divergent genomic and phenotypic profiles coexist and interact. Despite differing molecular backgrounds, those cell populations exhibit substantial differences in their metastatic ability, motility, proliferation, and differentiation potential [1]. Among them, a population of so-called cancer stem cells (CSCs) comes to the fore endowed with increased tumor-initiating capacities, self-renewal potential, and the ability to give rise to more differentiated progeny that forms tumor mass [2,3]. With their high metastatic potential and resistance, CSCs are responsible for tumor spreading, facilitating tumor relapse after treatment.

For the last several decades, enormous efforts have been undertaken to precisely characterize CSCs. However, it appeared extremely difficult as CSCs experience phases of transition between stem-like and non-stem-like states. Their phenotypical and functional properties are dynamically changing in response to genetic alterations, epigenetic factors, microenvironmental cues, or therapeutic agents, further adding to the complexity of cancer architecture [4].

Epigenetic mechanisms involving DNA methylation and chromatin modifications have emerged as crucial facilitators of cancer heterogeneity, substantially affecting cancer development and progression, modulating cell phenotypes, and either enhancing or inhibiting cancer cell malignant properties [5]. Chromatin structure may be altered transiently or persistently by several distinct molecular events, including DNA methylation, histone post-translational modifications (PTMs), or nucleosome positioning [6]. These alterations restrict chromatin accessibility, facilitating either chromatin condensation (formation of transcriptionally inactive heterochromatin) or chromatin relaxation (actively transcribed euchromatin); thereby substantially moderating the gene expression pattern [7]. The inherent reversibility of epigenetic mechanisms affecting transcriptional profiles of cancer cells facilitates the transition between distinct functional states, supporting intratumoral heterogeneity.

In this review, we have characterized the epigenetic mechanisms that regulate the acquisition and maintenance of cancer stemness concerning selected epigenetic factors belonging to the Bromodomain (BrD) family of proteins [8]. Bromodomain proteins are involved in a diverse range of functions, such as recognizing acetylated lysine on nucleosomal histones, remodeling chromatin, and recruiting other factors necessary for transcription [9]. These proteins thus play a critical role in the regulation of transcription, and recent reports suggest their essential role in mediating cancer stemness. Here, we reviewed the engagement of distinct BrD proteins in modulating cancer stem cell populations and highlighted their potential role as anti-tumor therapeutic targets, especially in stem cell-like cancers.

## 2. Cancer Stemness In Vitro and In Vivo

The stemness of cancer cells is a crucial feature for cancer progression, encompassing enhanced capacity for self-renewal, giving rise to differentiated cells, and interacting with the environment to maintain the balance between proliferation and quiescence [10]. To maintain the stem cell-like state, cancer cells frequently harness the mechanisms previously recognized as essential for normal stem cell self-renewal, including the induction of specific signaling pathways, the activation of core pluripotency transcription factor machinery, metabolic reprogramming, or epithelial-to-mesenchymal transition (EMT) [2].

JAK/STAT, Hedgehog, Wnt/β-catenin, Notch, PI3K/Akt, TGF-β, and NF-κB signaling pathways have all been shown to facilitate various stem cell characteristics in normal stem cell development. In many tumor types, the abnormal activation of those pathways supports cancer progression by contributing to the survival of the cancer stem cell-like population (reviewed in [11]). Also, the activation of core pluripotency transcription factor machinery [12,13,14], namely OCT4 (encoded by POU5F1), SOX2, Myc, or NANOG, is frequently observed in cancer stem cell-like tumor types and associated with worse patient survival. Poorly differentiated and highly aggressive cancers exhibit a gene expression profile mirroring the one observed for embryonic stem cells [12,13,15]. As OCT4, SOX2, Myc, and NANOG transcription factors are essential for normal stem cell propagation, they may also support the existence of cancer stem cells. Furthermore, a unique metabolic phenotype has emerged as critical for CSCs to sustain the stemness status. In response to tumor microenvironmental cues, CSCs exhibit metabolic plasticity between high levels of glycolysis and OXPHOS, which subsequently enables stemness maintenance [16,17]. The metabolic reprogramming supports the adaptation of CSCs to divergent exogenous stimuli without the loss of self-renewal potential. Also, tumorigenesis can trigger differentiated cancer cells to acquire a multipotent stem cell-like phenotype through EMT induction [18,19]. As the EMT program is regulated by the aforementioned developmental signaling pathways (also harnessed by CSCs) and serves as one of the mechanisms for switching phenotypes, it might also facilitate CSCs’ plasticity [20]. Therefore, CSCs exhibit more aggressive phenotype, enhanced motility, invasiveness, and great adaptation potency to microenvironmental signals; and self-renewal potency is regulated on many levels.

Many efforts have been taken to unequivocally define the CSC populations in distinct solid tumor types, resulting in the identification of many cell surface markers (i.e., ALDH1A1, CD24, CD44, CD49f, CD133, CD166, CD271, EpCAM, and others; reviewed in [14]), that enables for separation of CSCs in vitro. However, due to the CSC plasticity and differentiation potential, the obtained population will convert into cancer cells with non-stem cell-like phenotypes over time when cultured in vitro. The previous study clearly demonstrated that cancer cells could undergo a dynamic transition from a non-CSC to stem cell-like phenotype (and vice versa) to adapt to distinct stimuli in the tumor microenvironment [4].

To maintain the cancer stemness in vitro, anchorage-independent growth (three-dimensional cell cultures) assays are frequently applied, as theoretically, both self-renewal and differentiation can be evaluated at the single-cell level [21,22]. Sphere-forming cultivation of cancer cells also enriches the population in CSCs. The primary limitations of standard sphere-formation assays with a hanging drop approach or non-adherent culture plates include challenges in efficiently assessing the number and size of cultured spheres, as they are mobile and can undesirably merge. Fortunately, the semi-solid medium-based culturing systems overcome the previously mentioned limitations of sphere formation in suspensions. The utilization of a semi-solid medium prevents the migration and fusion of spheres and simplifies sphere counting and measuring [23].

CSCs are endowed with a tumor-initiating capacity, and when injected subcutaneously into mice models, results in the formation of tumors (xenografts) that consist of CSCs and more differentiated progeny [24]. This in vivo assay assumes that only CSCs can induce tumor formation and regenerate the heterogeneous population of cancer cells by undergoing asymmetrical divisions. Relying on the same premise, the limiting dilution assay (LDA) in vivo enables the estimation of the hypothetical frequency of CSCs in the population of cancer cells injected subcutaneously into mice models. The term limiting dilution is derived from the assay’s primary readout, which identifies the lower limit for the number of cells required for tumor initiation, supporting the stem-like property of CSCs [25].

Together with investigating the ability for migration, invasion, and the assessment of the CSC markers’ expression, the sphere formation and xenotransplantation assays constitute the golden standard for quantifying cancer stemness both in vitro and in vivo.

## 3. The Heterogeneity of Bromodomain Proteins

The Bromodomain (BrD) family of proteins encompasses 42 distinct members that possess at least one bromodomain—an evolutionarily highly conserved protein-interaction module that recognizes ε-N-lysine acetylation (Kac) motifs [26]. BrD consists of 110 amino acids in the form of four alpha helices αZ, αA, αB, and αC that are linked to each other by loop segments of variable length (ZA and BC loops). BrD selectively binds to Kac residues, particularly in histone proteins, although other non-histone protein targets of BrD proteins were also found. The hydrophobic nature of the Kac-binding pocket of the BrD and the fact that the strength of this BrD-Kac interaction is relatively low make these domains, particularly targetable by small molecules. In several BrD members, the BrD modules are present in multiples (i.e., PBR1 encodes 6 BrDs), while in most BrD proteins, the bromodomain is linked via flexible sequences to diverse interaction or catalytic domains resulting in a high divergence of a BrD protein family [8,26,27,28].

Bromodomain proteins were the first histone modification “readers” identified, displaying distinct specificity of histone recognition pattern. Therefore, BrD plays a critical role in orchestrating protein and DNA complexes at chromatin [8,26]. They represent a wide variety of functions in chromatin biology and gene transcription, including (i) acting as scaffold proteins that target chromatin-modifying enzymes and other molecular partners to specific sites in the chromatin, (ii) functioning as transcription factors or transcriptional co-regulators, (iii) or displaying various catalytic functions, including methyltransferase or acetyltransferase activities. Their intact activity provides proper control of cell epigenetic identity. When altered, BrD members may contribute to the transcriptional deregulation that ultimately results in the acquisition of cancer hallmarks [8,26,27,28,29].

Based on their functional divergence, Zaware et al. [8] have proposed the classification of BrD proteins into 9 groups, including (I) histone acetyltransferases (9 members), (II) histone methyltransferases (2 members), (III) chromatin remodeling factors (11 members), (IV) AAA ATPase proteins (2 members), (V) BET family transcriptional co-activators (4 members), (VI) E3 SUMO/ubiquitin ligases (4 members), (VII) SP family proteins of PML nuclear bodies (4 members), (VIII) transcriptional co-repressors (2 members), and (IX) WD-repeat proteins (3 members). This function-based organization presents some overlap with the conventional structure-based classification [26], although it demonstrates more clearly the dependence of BrD proteins on other structurally conserved modular domains they possess and utilize to exert effects on protein-protein or protein-nucleic acid interactions.

There are five BrD proteins that possess the histone acetyltransferase (HAT) activity, including KAT2A (also known as GCN5), PCAF (KAT2B), p300/CBP (EP300), TAF1 (TBP-associated factor 1, also known as TAFII250) and TAF1L, in which the BrD and HAT catalytic domain are in close proximity. These birds are classified as group Ia [8,30]. The bromodomains in these nuclear HATs contribute to the substrate recruitment and specificity involving histones and non-histone proteins, thereby providing a functional link between lysine acetylation and acetylation-mediated protein-protein interactions in chromatin-mediated gene transcription [31]. Also, three of the BrD proteins, namely BRPF1, BRPF2 (BRD1), and BRPF3, act as scaffold subunits of various HAT complexes (such as the MOZ/MORF and HBO1), stimulating acetyltransferase and transcriptional activity of the complexes. These proteins are not endowed with the HAT catalytic activity itself. Together with BRD8, which is a component of the NuA4 histone acetyltransferase complex, they form the group Ib of bromodomain proteins that mediates histone acetylation [8,26,30].

Albeit KMT2A (also known as MLL) and ASH1L possess bromodomains, they are mostly known as SET domain-containing histone methyltransferases (HMTs) associated with transcriptionally active chromatin sites [31]. The bromodomain of KMT2A does not bind acetylated lysine residues; however, it plays an essential role in modulating functions of the adjacent PHD3 finger, influencing its interaction with tri-methylated lysine 4 of histone H3. As a part of the chromatin remodeling machinery, KMT2A predominantly forms H3K4me1 and H3K4me2 methylation marks, while the ASH1L deposits the H3K36me3 methylation mark. The bromodomain of ASH1L lacks the conserved Asn residue required for Kac binding, lies C-terminally to the SET domain, and its role remains not poorly understood [26,30,31].

The III group of bromodomain proteins includes (i) the members of the SWI/SNF family of chromatin remodeling complexes (group IIIa), namely the SMARCA2 (BRM), SMARCA4 (BRG1), BRD7, BRD9, and PBRM1; and (ii) the members of the ISWI family of chromatin remodeling complexes (group IIIb), namely BAZ1A, BAZ1B, BAZ2A, BAZ2B, BPTF, and CECR2 [8]. The chromatin remodelers are huge multiprotein complexes that use the energy from ATP hydrolysis to maintain correct gene expression profiles, chromatin stability, DNA repair, DNA replication, and inherited epigenetic states. Based on their ATPase subunit, they are categorized into four large families: SWI/SNF, ISWI, INO80, and CHD [32].

In SWI/SNF complexes, the SMARCA4 (BRG1) or SMARCA2 (BRM) bromodomain proteins, which are mutually exclusive, possess the ATPase activity, and their bromodomains could contribute to either assembly or targeting of the SWI/SNF complex to specific genomic loci [33]. A recent study revealed the modular organization and assembly of three distinct classes of SWI/SNF complexes, including canonical BRG1/BRM-associated factor (cBAF); polybromo-associated BAF (PBAF), with PBRM1 and BRD7 as additional bromodomain components; and a newly defined non-canonical BAF (ncBAF) complex, containing BRD9 [34]. Except for the PBRM1 protein, which possesses 6 bromodomain modules, all group III bromodomain proteins encode only one BrD unit [8,26].

The ISWI complexes are one of the best-conserved ATPase families [35]. They harbor either SMARCA1 (SNF2L) or SMARCA5 (SNF2) as ATPase subunits and one to three distinct non-catalytic subunits, with six different bromodomain proteins among them: BAZ1A, which is present in ACF or CHRAC complexes, BAZ1B—in WICH complexes, BAZ2A—in NRC complexes, BAZ2B—in BRF complexes, BPTF—in NURF complexes, and CECR2—in CERF complexes [8,26,30]. The inherence of bromodomain proteins contributes to the proper functions of ISWI complexes, including chromatin assembly and replication mediated by ACF, CHRAC, or WICH complexes or transcriptional regulation mediated by NURF or NRC complexes. Generally, ISWI complexes support the maturation of initial histone–DNA complexes (pre-nucleosomes) into canonical octameric nucleosomes and the spacing of nucleosomes at relatively fixed distances [35]. Also, ISWI complexes are involved in multiple aspects of cell physiology, such as transcriptional regulation [36]; and DNA damage response, repair, and recombination [37,38]. ISWI actions in cancer are gene- or context-dependent, and the interaction with distinct transcription factors may establish different tumor properties.

ATAD2 (ANCCA) and its paralog ATAD2B (KIA1240) are the ATPase family members that contain the ATPases associated with diverse cellular activities (AAA) domains, as well as the bromodomain, and according to the functional classification proposed by Zaware et al. [8], both proteins are classified as a group IV bromodomain protein. ATAD2 was previously identified as a nuclear co-activator for estrogen (ER) and androgen (AR) receptors regulating the hormone-induced expression of genes involved in the proliferation and survival of cancer cells [39]. Also, ATAD2 interacts with Myc, E2F, and SOX10 transcription factors facilitating cancer progression [40,41]. Through the BrD domain, ATAD2 recognizes acetylated lysine 5 on histone 4 (H4K5ac) and lysine 12 on histone 4 (H4K12ac), and associates with di-acetylated histone H4K5acK12ac modifications found on newly synthesized histones following DNA replication. Similarly, ATAD2B bromodomain is a diacetyllysine reader module; however, the significance of histone H4K5acK8ac ligand binding is yet to be determined [42].

The V group of bromodomain proteins comprises BRD2, BRD3, BRD4, and BRDT proteins, which all possess two tandem bromodomains and an extra terminal (ET) and are known as the BET family [43]. Except for BRDT, which is specifically present in the testis and ovaries, all other BET members are ubiquitously expressed in normal tissues [26]. As epigenetic readers, BET proteins exhibit broad specificity on transcriptional activation, recruiting transcriptional regulatory complexes to acetylated chromatin and supporting transcription factors stability. BET proteins facilitate RNA Pol II pause-release and transcriptional elongation, and are also engaged in chromatin remodeling, DNA damage response and genome integrity, and super-enhancer assembly [26,44]. In contrast to other BrD proteins, BET subfamily members prefer to bind to di-acetylated lysine residues closely located in the protein sequence, recognizing acetylated both histone 3 and histone 4. Previously, genetic screening in distinct types of tumors has identified BET family members as indisputable for cancer cell survival. Specifically, BRD4 was proposed as a druggable target for Myc-driven tumors [45], although the potential of other BET proteins should not be omitted.

The VI group of BrD proteins that possess both the SUMO and ubiquitin E3 ligase activities is formed by four proteins collectively known as transcriptional intermediary factor 1 (TIF1) chromatin-binding proteins, namely TIF1α (TRIM24), TIF1β (TRIM28) TIF1γ (TRIM33), and TIF1δ (TRIM66) [46]. All members of the TIF1 family encode a single BrD domain in tandem with the plant homeodomain (PHD) domain at the C-terminus of the polypeptide. The BrD-PHD unit is necessary for interacting with modified histones, robustly contributing to the maintenance of genome stability [8,26]. Also, TIF1 proteins harbor N-terminally encoded tripartite motif (TRIM) composed of the RING domain, two B-boxes, and a coiled-coil domain; and therefore, they are also classified as members of the TRIM family of E3 ubiquitin ligases [47]. While TIF1 members exhibit high structural homology, they exert diverse functions in normal and cancer cell biology, being involved in the regulation of genomic stability, chromatin compaction, the DNA damage response pathway, regulation of the cell cycle progression, cellular metabolism, and plasticity [47].

The VII group of BrD proteins encompasses four highly similar speckled proteins (SP)—SP100, SP110, SP140, and SP140-like protein (SP140L) [8,26]. They are associated with promyelocytic leukemia nuclear bodies (PML-NBs), multiprotein complexes present in a variety of diseases, including cancer. SP family members share a similar structure, encoding three functional domains: the SP100, Aire, NucP41/P75 and Deaf (SAND) domain, PHD, BrD, and caspase activation and recruitment (CARD) domain. The SAND domain is associated with chromatin-dependent transcriptional regulation through direct DNA binding [48], while PHD and BrD create a dual-reader module, where PHD recognizes unmethylated lysine in H3 (H3K4me0), and BrD stabilizes PHD fold and reads histone acetylation [49]. The CARD domain can induce SP homo- or heteromultimerization. The SP family has recently received a lot of interest for its role in chromatin and transcriptional regulation in immune cells during homeostasis, immunity to bacterial and viral infections, and the development of inflammatory diseases or immunodeficiency [50]. However, their role in cancerogenesis is scarcely documented.

Two well-known transcription regulators, ZMYND8 (RACK7) and ZMYND11, constitute the VIII group of BrD proteins [8]. Both proteins possess an N-terminally encoded bromodomain in a tripartite PHD-BrD-PWWP module, which enables recognition of several distinct post-translational modifications on histone proteins, including dual signature H3K4me1/H3K14ac by ZMYND8 and H3.3K36me3 by ZMYND11, respectively [26,30,51,52]. Previous studies revealed that ZMYND8 and ZMYND11 might act as either transcriptional activators or repressors, and both proteins harbor tumor suppressor functions, particularly in breast, prostate, colon, and ovarian carcinoma [51,52]. Also, ZMYND8 was recently reported as a critical DNA damage response factor involved in regulating transcriptional responses and DNA repair activities at DNA double-strand breaks [51].

The IX group of BrD domains is formed by three WD-repeat proteins, namely BRWD1 (WDR9), BRWD3, and PHIP (WDR11), that possess two BrD units and several WD repeats [8]. They share a high structural homology. Previous reports revealed their engagement in chromatin regulation, cell cycle progression, signal transduction, or acting as scaffold proteins for other factors [26,30].

## 4. Several BrD Family Members Play a Fundamental Role in Cancer Stem Cell Maintenance

Epigenetic mechanisms that encompass diverse post-translational histone modifications, DNA methylation, chromatin remodeling, and even changes in non-coding RNAs, altogether dictate the outcome of cell fate specification. Not surprisingly, considering the importance of epigenetic regulators in normal stem cell maintenance, many chromatin-related proteins are essential to maintaining the CSC state [7,17]. Indeed, we have recently reported significant positive associations of selected BrD proteins with cancer de-differentiation status, regardless of the tumor type. Here, we would like to discuss the current state of the art considering bromodomain proteins and their role in regulating the cancer stem cell-like phenotype of solid tumors.

To date, several BrD family members (KAT2A, EP300, TAF1, BRPF1, KMT2A, SMARCA4, BAZ1B, BAZ2A, ATAD2, BRD4, TRIM24, TRIM28, and ZMYND8) were recognized and confirmed (mechanistically, at the molecular level) as positive regulators/mediators of cancer stemness, while for the SMARCA2 protein—mostly negative roles in CSC maintenance were reported (Figure 1). However, the engagement of the majority of BrD proteins in mechanisms regulating cancer de-differentiation status still remains undiscovered or not fully specified, implying rather a positive role in maintaining stemness. Also, several BrD proteins (including BAZ1A, ATAD2B, SP100, SP110, SP140, SP140L, BRWD1, BRWD3, and PHIP) are poorly studied, and little is known about their role in oncogenesis, leaving space for identifying more stemness-promoting or stemness-inhibiting proteins among the BrD family. Nevertheless, the most prominent functions of BrD proteins in regard to cancer de-differentiation status are presented below.

### 4.1. BrD Proteins with Histone Acetyltransferase (HAT) Activities and Cancer Stemness

KAT2A (GCN5) and KAT2B (PCAF)

KAT2A and KAT2B are two highly homologous orthologues that exhibit a mutually exclusive pattern of expression. While KAT2A dominates in embryonal development, neural tissue differentiation, and hematopoiesis, KAT2B prevails in skeletal muscles. Only KAT2A is essential for proper embryonic development, stabilizing pluripotency gene regulatory networks and acting as a co-factor of the Myc transcription factor [53,54]. Conversely, KAT2B is minimally expressed at the early stages of development and upregulated in adult tissues. Recent studies demonstrated the significant contribution of KAT2A to cancer progression. Specifically, KAT2A upregulation associates with worse patient outcomes in several cancers, including breast, lung, colon, and renal cancers [55,56,57,58]. KAT2A contributes to cancer through the control of transcriptional activity, mainly the co-activation of E2F and Myc transcriptional targets [59]. In renal cancer, KAT2A promotes tumor growth and accelerates the metastatic phenotype of cancer cells by regulating glycolytic metabolism through MCT1 upregulation [60]. A recent study demonstrated that loss of KAT2A enhances transcriptional noise and abrogates acute myeloid leukemia stem-like cells [61]. KAT2A is also a critical component of a well-known stemness-associated TGF-β/SMAD signaling pathway, working downstream of TGF-β/SMAD to regulate the epithelial-to-mesenchymal transition in breast cancer [58]. We have recently shown that KAT2A high expression significantly associates with cancer stemness across distinct types of solid tumors, although mechanistic data in support of this hypothesis is currently missing. Correspondingly, Martile et al. [62] have demonstrated that KAT2A inhibition reduced the viability of lung cancer stem-like cells.

In contrast, KAT2B overexpression correlates negatively with cancer de-differentiation status, which is in line with the previously reported role of KAT2B in mediating differentiation of normal multipotent progenitor cells [63]. However, KAT2B was shown as a positive co-factor of the Hedgehog–Gli signaling pathway in brain tumors, and KAT2B silencing attenuated the tumor-forming capacity of neural stem cells in vivo [64], suggesting that the KAT2B role in stem cell biology might be highly context-dependent.

P300/CBP (EP300)

EP300 encodes a multi-domain protein that functions as acetyltransferase for both histone and non-histone targets. In somatic cell-induced reprogramming, p300 promotes acetylation of OCT4, SOX2, and KLF4 at multiple sites to change their transcription activity, thus regulating stemness acquisition [65]. In cancer, EP300 has been demonstrated as an oncogene promoting tumor growth and metastatic potential, and facilitating cancer stemness in breast cancer. Notably, Mahmud et al. [66] have shown that EP300 upregulation in vitro in the triple-negative breast cancer (TNBC) cell line led to an increased expression of mesenchymal and stem cell markers, enhanced migration, invasion, anchorage-independent growth, and enriched drug resistance. Cho et al. [67] reported that the formation of c-Myc-p300 complex, which further cooperates with DOT1L, is critical for the regulation of the EMT and EMT-associated CSCs in breast tumor initiation and progression. The engagement of EP300 in EMT was further observed in nasopharyngeal carcinoma, where p300 promoted EMT through the acetylation of Smad2 and Smad3 in the TGF-β signaling pathway [68]. Also, Ring et al. [69] demonstrated that EP300 knockdown reduced the CSC population in TNBC. Correspondingly, p300 promotes cell proliferation, migration, and invasion via inducing EMT in non-small cell lung cancer (NSCLC) cells [70]. In hepatocellular cancer, high expression of p300 was correlated with enhanced vascular invasion, increased metastatic potential, and worse survival [71,72,73]. In glioma, P300 mediates glioma stem cell adaptive response to therapeutic stress [74]. On the other hand, in pancreatic cancer, the p300/GATA6 axis determines differentiation and Wnt dependency, and loss of EP300 leads to a phenotypic transition from the classical subtype to the de-differentiated basal-like/squamous subtype of pancreatic cancer due to the attenuation of the GATA6-regulated differentiation program [75]. Similarly, Asaduzzaman et al. [76] have demonstrated that the modulation of EP300 expression alters cancer stem cell markers and anchorage independence in basal-like breast cancer models. EP300 downregulation resulted in a more malignant phenotype of breast tumors with the acquisition of drug resistance, although the mechanistic explanation was not presented in this study. Therefore, the p300 activity in the stem cell-like compartment of cancer might result from specific functions of its binding partners, mostly transcription factors, either exerting stemness-supporting or differentiating transcriptional programs.

TAF1 and TAF1L

The TAF1 gene encodes the largest subunit of the transcription factor II D (TFIID) in the RNA Polymerase II initiation complex, which promotes transcriptional initiation and activation [77]. The TAF1 homolog TAF1L has 95% amino acid identity with TAF1 protein, and both proteins are potent druggable bromodomain family members [78]. As with many other TAFs, TAF1 mediates activator-dependent transcription in a promoter- and tissue-specific manner [79]. To date, only several studies have analyzed the roles of TAF1 or TAF1L in facilitating cancer development and progression. Notably, Zhang et al. [80] have recently demonstrated that TAF1 promotes EMT in non-small cell lung cancer by transcriptionally activating TGFβ1. Higher TAF1 expression is associated with worse outcomes in NSCLC patients, and the authors proposed TAF1/TGFβ1 as a novel therapeutic target. In glioma, TAF1 is a direct binding partner of long non-coding RNA (lncRNA) LIN00319, modulating the tumorigenesis by the upregulation of the HMGA2 oncogene [81]. Moreover, the lncRNA FOXD2-AS1 recruits TAF1 to the NOTCH1 promoter, inducing the NOTCH signaling pathway, and thereby promoting stemness, while impairing cell apoptosis and differentiation of glioma cancer cells [82]. As for TAF1L, the overexpression promotes cell proliferation, migration, and invasion in esophageal squamous cell carcinoma (ESCC) by modulating the Akt signaling pathway [83]. Moreover, TAF1L enables oral squamous cell carcinoma (OSCC) cells to avoid apoptosis by activating autophagy [84]. Therefore, a high TAF1L level enhances OSCC development, although a direct mechanistic link to cancer stemness is still missing.

### 4.2. BrD Proteins as Scaffold Proteins for HAT Complexes and Cancer Stemness

BRPF1, BRPF2, and BRPF3

BRPF1 is a multivalent chromatin regulator, encoding several modules that recognize post-translational modifications on histone proteins, including double PHD fingers (targeting unmethylated histone H3), a bromodomain (preferentially interacts with H2AK5ac, H4K12ac, and H3K14ac), and a PWWP domain (recognizes H3K36me3 mark). BRPF2 and BPRF3 are paralogs of BRPF1 with conserved domain architecture [8,26].

As a scaffold protein, BRPF1 interacts with the lysine acetyltransferases KAT6A, KAT6B, and KAT7, also known as MOZ, MORF, and HBO1, respectively. BRPF1 promotes the quartet-complex formation, restricts substrate specificity, and enhances the enzymatic activity of those acetyltransferases. BRPF2 preferentially associates with HBO1, assembling a chromatin complex required for the global acetylation of H3K14ac [85].

Previously, You et al. [86] uncovered a crucial role of Brpf1 in controlling mouse embryo development and regulating cellular and gene expression programs. Also, knockout of Brpf2 leads to embryonic lethality at E15.5 due to its role in ESC differentiation [87]. In contrast, Brpf3 was demonstrated as not necessary for proper mouse embryo development and survival, distinguishing BRPF3 from its paralogs [88]. However, this was recently denied by Cho et al. [89] who revealed the specific function of Brpf3 in proper differentiation, as well as the cell-cycle progression of ESCs via regulation of Myst2 acetyltransferase stability. As all BRPF paralogs are involved in normal stem cell regulation, they may also be important contributors to cancer development. In fact, in glioma, BRPF1 regulates cancer cell proliferation and colony formation and was recognized as a potential therapeutic target for primary LGG [90]. Similarly, Cheng et al. [91] demonstrated that BRPF1 might serve as a druggable target in liver cancer, where BRPF1 was involved in the regulation of cell cycle progression, senescence, and cancer stemness of hepatocellular carcinoma. In prostate tumors, accumulated BRPF1 protein accelerated the cell growth, stem-like properties, and migration of cancer cells, further supporting the role of BRPF1 in cancer stemness [92]. As for BRPF2 and BRPF3, little is known about their engagement in cancer development and progression, leaving an open question of whether they exert similar activity as BRPF1 in maintaining cancer stem cell-like phenotype.

BRD8

BRD8 has been recognized as a non-catalytic component of the conserved nucleosome acetyltransferase of the H4 (NuA4) complex, which is a multisubunit HAT mediating the acetylation of the N-terminal tail of histones H4 and H2A. The previous report demonstrated that BRD8 expression is associated with colorectal tumor progression toward advanced stages and may aid in gaining a growth advantage [93]. BRD8 downregulation significantly impairs cancer cell proliferation of colorectal and hepatocellular tumors, albeit there is no direct evidence of modulating the cancer stem cell-like compartment of the tumor [94,95].

### 4.3. BrD Proteins with Histone Methyltransferase (HMT) Activities and Cancer Stemness

KMT2A

One of the bromodomain proteins that possess intrinsic histone methyltransferase activity is an MLL1 protein encoded by KMT2A. MLL1 enzyme trimethylates H3K4 via its SET domain, mediating transcriptional activation [8,30]. The bromodomain of MLL1 protein lacks the conserved asparagine (involved in acetyllysine recognition), and enhances the interaction of the adjacent PHD domain with the H3K4me3 histone mark [96]. MLL1 functions as a part of a multiprotein complex that includes RBBP5, WDR5, and ASH2L [97]. MLL1 has an essential role in regulating the expression of genes that are implicated in the self-renewal of hematopoietic stem cells and unequivocally contribute to leukemia development. In embryonal development, homozygous deletion of KMT2A is lethal [98]. MLL1, together with SET1, NANOG, and BACH1 supports the pluripotency of mouse ESCs by maintaining the H3K4me3 state and enhancer-promoter activity, especially on stemness-related genes [99].

KMT2A was also demonstrated as essential for mediating cancer stemness in solid tumors. Grant et al. [100] have recently reported that the ablation of MLL1 decreases the self-renewal of human colon cancer spheres and halts tumor growth in vivo. Mechanistically, MLL1 controls the expression of stem cell genes, including the Wnt/β-catenin target gene, Lgr5. Similarly, MLL1 acted synergistically with b-catenin in cervical carcinoma, promoting tumorigenesis and metastasis [101]. Using a panel of cancer cell lines from distinct solid tumors, Ansari et al. [102] revealed that MLL is essential for cell survival, tumor growth, hypoxia signaling, and angiogenesis. In melanoma cell lines, KMT2A knockdown markedly decreased the expression of cancer stem cell markers, namely Nanog, Oct-4, and Sox-2, and significantly repressed the tumorsphere formation ability. Zhang et al. [103] proposed that KMT2A regulated melanoma cell growth by activating the hTERT-dependent signal pathway and suggested the KMT2A/hTERT axis as a potential therapeutic target. Taken together, KMT2A/MLL1 was identified as a positive regulator of cancer stemness.

ASH1L

ASH1L encodes another bromodomain–containing HMT that can associate with actively transcribed loci, mediating dimethylation of H3K36 through its’ SET domain. An essential role for ASH1L was previously established in regulating normal stem cell maintenance, including hematopoietic stem cells and mesenchymal stem cells [104]. ASH1L was found to be upregulated in thyroid, breast, and liver cancers and was linked to enhanced cancer cell growth and aggressiveness of the tumor [105,106,107,108]. In breast cancer, higher ASH1L levels were observed in the basal subtype, the one that exhibits cancer stem cell-like traits [109]. To date, molecular studies confirming ASH1L involvement in the regulation of cancer de-differentiation status in solid tumors are missing.

### 4.4. BrD-Encoding Members of the SWI/SNF Family of Chromatin Remodeling Complexes and Cancer Stemness

SMARCA2 and SMARCA4

SMARCA2 (BRM) or SMARCA4 (BRG1), both of which contain a C-terminal BRD module that can recognize acetylated histones H3 and H4 [8,26], act as core subunits of the SWI/SNF chromatin remodeling complexes, mediating ATP-dependent alteration of chromatin structure. Despite sharing high structural homology, SMARCA2 and SMARCA4 play distinct biological functions and exhibit diverse expression patterns [110,111]. In embryonic stem cells, deletion of SMARCA4 led to the loss of self-renewal, and SMARCA4 null mutants caused early embryonic lethality, while SMARCA2-knockout mice lived until adulthood [112].

In cancer, both SMARCA2 and SMARCA4 have emerged as critical tumor suppressors [113,114], although several reports unveil their oncogenic potential [115,116]. Also, high expression of SMARCA4 or SMARCA2 is frequently associated with an opposite prognosis in cancer, and their levels correlate inversely with the histologic tumor grade [117].

Recently, we have demonstrated that SMARCA4 exhibit a consistent positive correlation with cancer stemness (assessed based on transcriptomic features of TCGA cancers). At the same time, SMARCA2—shows a negative association with tumor de-differentiation status, regardless of the tumor type [118]. Indeed, the fundamental role of SMARCA4 in the regulation of cancer stemness was previously reported by Yoshikawa et al. [119,120] in colorectal cancers. SMARCA4 was determined as essential for maintaining the stem cell-like traits of intestinal tumor cells cultured in spheroid systems in vitro. SMARCA4 knockdown robustly impaired cell proliferation and increased apoptosis of cancer cells. Also, SMARCA4 is a crucial regulator of CD44 expression—the most commonly used marker of CSCs—that facilitates tumor invasion and metastasis. Yan et al. [121] have demonstrated that SMARCA4 interacts with RUNX2 and together promotes the CD44-induced EMT in colorectal carcinoma. In hepatocellular carcinoma, SMARCA4 activates a series of downstream cascades by directly binding to the Sall4 promoter and enhancing Sall4 transcription, ultimately resulting in enhancing the stemness potency of cancer cells [122]. SMARCA4 also plays a fundamental role in maintaining glioma-initiating cells that present stem-like molecular features [123]. SMARCA4-containing BAF complex maintains glioma stem cells in a cycling, oligodendrocyte precursor cell (OPC)-like state, and disruption of SMARCA4 promotes the progression of differentiation along the astrocytic lineage [124]. On the contrary, SMARCA4 interacts with TGFB2-AS1 long non-coding RNA in breast cancer, resulting in transcriptional repression of TGFB2 and SOX2, and therefore, leading to attenuation of TGF-β signaling and loss of cancer stem cell-like characteristics [125]. Furthermore, loss of SMARCA4 in lung adenocarcinoma resulted in the acquisition of highly de-differentiated cancer cell traits and increased metastatic incidence, suggesting that the involvement of SMARCA4 in regulating cancer stemness might be associated with the cell of origin [126].

The available data concerning SMARCA2 in cancer suggest that SMARCA2 function differs depending on the cancer type. In some tumor types, namely lung, kidney, and breast tumors, the event of SMARCA2 downregulation occurs during cancer de-differentiation (disease progression), suggesting a clonal selection of SMARCA2-abrogated cancer cells [127,128,129,130]. In breast cancer cell lines in vitro, the overexpression of SMARCA2 was suppressed, whereas SMARCA2 knockdown promoted TGF-β-induced migration and invasion of cancer cells [130].

On the other hand, in pancreatic cancer, the upregulation of SMARCA2 associates with disease progression, being significantly positively correlated with patients’ poor survival, larger tumor size, metastases, lymphatic invasion, and stage IV disease [131]. We have recently reported a consistent negative association of SMARCA2 expression with cancer de-differentiation status [118]. However, the molecular studies demonstrating the precise role of SMARCA2 in the regulation of cancer stemness across distinct types of solid tumors are insufficient to make an unequivocal decision on whether SMARCA2 is a negative cancer stemness regulator.

BRD7 and PBRM1

BRD7 and PBRM1 are additional bromodomain components of the polybromo-associated BAF (PBAF) class of SWI/SNF complexes, engaged in transcriptional regulation, DNA repair, and regulation of chromatin architecture and topology [132]. Furthermore, BRD7 interacts with numerous transcription factors, playing crucial roles in cell proliferation, apoptosis, differentiation, and glucose metabolism [133,134,135,136]. As for PBRM1, molecular studies revealed its engagement in facilitating genome stability, in the repair of the DNA double-strand breaks and ubiquitinating PCNA [137].

In embryonic stem cells, BRD7 is required for either activation or repression of specific genes, regulating stem cell self-renewal [110]. Inactivation of PBRM1 in mice leads to embryonic lethality at E11.5 due to heart defects [138].

An increasing number of studies have found that BRD7 expression is decreased or lost in human cancers, which is in line with its role as a tumor suppressor [139]. Previous studies demonstrated that BRD7 delays tumor progression by negatively regulating the PI3K/AKT, P53, Ras-Raf-MEK-ERK, and β-catenin pathways [140,141,142,143], and a significant BRD7 downregulation was observed in breast, colon, lung, ovarian, and endometrial cancers [134,139,144,145,146].

On the other hand, Zhao et al. [147] demonstrated that in colorectal cancer, BRD7 exerts oncogenic activity, which did not arise from its function as an SWI/SNF subunit. BRD7 stabilizes the level of c-Myc by decreasing its ubiquitination and proteasomal degradation. Furthermore, a high level of BRD7 is positively associated with c-Myc expression, clinical stage, and poor prognosis in colorectal cancer patients, further supporting the role of BRF7 in Myc-dependent tumors [147]. However, it does not clarify whether BRF7 is essential for maintaining cancer stemness.

In cancer, PBRM1 acts primarily as a tumor suppressor, and loss of PBRM1 was previously associated with the deregulation of expression of apoptotic and cell cycle regulatory genes [137], as well as the metabolism-related markers and cell adhesion molecules [148]. In embryo development, PBRM1 ablation results in heart defects and, ultimately, embryonic lethality at E15.5 [149]. This suggests that PBRM1 is necessary for the regulation of proper cell differentiation rather than sustaining stem cell self-renewal. To date, the involvement of PBRM1 in cancerogenesis has been mainly reported in kidney tumors [138,150,151,152]. However, these reports did not focus on the aspect of cancer de-differentiation. Recently, Hagiwara et al. [153] have demonstrated that an oncogenic protein, MUC1-C, that drives lineage plasticity in prostate cancer progression also integrates the activation of PBRM1 with the regulation of redox balance, pluripotency markers’ expression, and cancer stem cell-like state in pancreatic cancer. Our previous report demonstrated a negative association of PBRM1 expression with cancer stemness in several types of solid tumors, including kidney and pancreatic cancers, although mechanistic studies are missing [118].

BRD9

A non-canonical BRD9-containing BAF (ncBAF) chromatin remodeling complex maintains naive pluripotency in mouse ESCs, and inhibition of BRD9 results in transcriptional changes representative of a primed epiblast-like state [154]. Recently, BRD9 was also established as an essential factor that safeguards somatic cell identity, and BRD9 inhibition lowered chromatin-based barriers to reprogramming to pluripotency [155].

In cancer, degradation of BRD9-induced downregulation of oncogenic transcriptional programs inhibited tumor progression in vivo only in a synovial subtype of sarcomas (and synovial sarcoma is concerned with stem cell malignancy) [156,157]. Later, oncogenic functions of BRD9 were also reported in squamous cell lung carcinoma. Huang et al. [158] demonstrated that BRD9 inhibition remarkably reduced tumorigenesis by downregulating c-Myc activity. Furthermore, Bell et al. [159] highlighted the BRD9 as an indisputable facilitator of the c-Myc-related gene signature of mammary cells and their oncogenic potency. In hepatocellular carcinoma, BRD9 promoted the proliferation, migration, invasion, and EMT of cancer cells [160,161]. Fang et al. [161] determined the Wnt/β-catenin signaling pathway—a well-known mediator of stem cell maintenance—as directly activated by BRD9 in HCC, although they did not evaluate whether BRD9 is indispensable for sustaining cancer stemness. In kidney cancers, BRD9 promoted tumor malignancy by activating the Notch signaling pathway in HIF-2a tumors [162]. Also, BRD9 was reported as a diagnostic biomarker and drug target in metastatic prostate cancer [163] and a critical regulator of androgen receptor signaling, facilitating prostate cancer progression [157]. Zhu et al. [164] showed that BRD9 overexpression in colon cancer cells notably elevated proliferation and migration potencies. While all of the abovementioned studies did not focus directly on cancer stemness molecular traits, they allow the assumption of the stemness-supporting role for BRD9.

### 4.5. BrD-Encoding Members of the ISWI Family of Chromatin Remodeling Complexes and Cancer Stemness

BAZ1B, BAZ2A, and BAZ2B

BAZ1B (also known as WSTF, the component of the WICH complex) and BAZ2A (also known as TIP5, the component of the NRC complex) are relatively well-recognized in cancer-supporting programs. BAZ1B exerts oncogenic functions in breast, colorectal, and lung cancers [165,166,167]. Specifically, BAZ1B overexpression promotes proliferation, colony formation, migration, and invasion of lung cancer cells. Molecularly, BAZ1B activates both PI3K/Akt and IL-6/STAT3 oncogenic signaling pathways, which were previously linked to cancer stemness [167]. In our previous report, we determined a positive correlation between BAZ1B expression and cancer stemness in several types of solid tumors, including breast and lung cancers [118]. Therefore, BAZ1B might be concerned as a cancer stemness-promoting factor.

As for BAZ2A, we observed either a positive or negative association with cancer de-differentiation, which seemed to be tumor-specific. However, Peña-Hernández et al. [168] revealed that BAZ2A-mediated repression via H3K14ac-marked enhancers promotes prostate cancer stem cells. Not surprisingly, given that BAZ2A maintains ESC pluripotency by safeguarding the genome architecture [169]. Also, BAZ2A enhanced the tumorigenicity of hepatocellular carcinoma. Li et al. [170] demonstrated that upregulation of BAZ2A promoted the growth of HCC cells in an anchorage-independent growth assay (frequently used to analyze the stem cell-like properties of tumors) by inducing the Wnt/β-catenin signaling pathway. This further supports the role of BAZ2A in mediating cancer stemness.

BAZ2B (the component of the BRF complex) engagement in regulating stemness was previously demonstrated in hematopoietic cells. As shown by Arumugam et al. [171], BAZ2B overexpression reprogramed the hematopoietic lineage-committed progenitors into a multipotent stem state, resulting in reprogrammed cells with an increased long-term clonogenicity, enhanced engraftment potential and ability to differentiate into multiple lineages. However, the role of BAZ2B in regulating cancer development and progression remains elusive. In our report, we observed that BAZ2B correlates negatively with cancer stemness in distinct types of solid tumors [118], although it requires further study to determine the molecular mechanism of BAZ2B involvement in cancer de-differentiation.

BPTF and CECR2

In normal embryonic stem cells, the BPTF (the component of the NURF complex) is required for proper cell differentiation, and mutations in BPTF result in a lethal phenotype (E8.5) [172]. Also, BPTF activates a stemness gene-expression program essential for the maintenance of adult hematopoietic stem cells and maintains the self-renewal capacity of mammary gland stem cells [173].

BPTF was previously identified as a tumor-promoting factor in several cancer types, including lung, liver, ovarian, gastric, colorectal, and brain cancers [174,175,176,177,178,179,180,181]. In liver tumors, BPTF promotes tumor growth by modulating hTERT signaling and cancer stem cell traits. Specifically, BPTF knockdown inhibited cell proliferation and colony formation and resulted in the downregulation of stem cell markers expression (both in vitro and in vivo) [177]. Miao et al. [178] revealed that in ovarian carcinoma, BPTF downregulation inhibited cell proliferation, colony formation, migration, and invasiveness and induced apoptosis. Also, BPTF knockdown affected the EMT signaling pathway—a critical regulator of cancer stemness. Moreover, BPTF was recognized as a c-Myc interactor required for c-Myc chromatin recruitment and transcriptional activity in several types of tumors [180,181,182]. Therefore, BPTF inhibition has arisen as a promising strategy to combat cancer through epigenetic regulation of the c-Myc oncogenic pathway.

CECR2 (the component of the CERF complex) was recently reported to promote somatic cell reprogramming that worked through a protein network to overcome epigenetic barriers to induced stemness [183]. The molecular and clinical implications of CECR2 in cancer remain largely unknown, although a very recent report revealed that CECR2 is a driver for breast cancer metastasis by promoting NF-kB signaling and macrophage-mediated immune suppression [184]. Still, the mechanistic link to modulating cancer de-differentiation remains unresolved.

### 4.6. BrD Proteins with AAA+ ATPase Activity and Cancer Stemness

ATAD2

ATAD2 takes part in a range of cellular activities, such as transcriptional regulation, chromatin remodeling, histone modification, and the formation of higher-order chromatin structures [8,26,30]. ATAD2 is normally overexpressed in nonspecialized cells, including embryonic stem cells and germ cells. Previously, ATAD2 has been recognized as critical in supporting specific transcriptional programs in ESC cells, modulating their proliferation and differentiation.

To date, there are several reports implying ATAD2 involvement in mechanisms related to cancer stemness. Specifically, Li et al. [185] have recently demonstrated that the downregulation of ATAD2 restrains CSC-like properties in esophageal squamous cell carcinoma (ESCC) via blockade of the Hedgehog signaling pathway. Hao et al. [186] have observed that ATAD2 knockdown inhibited the migration, invasion, stem cell-like properties, and mitochondrial reactive oxygen species (mtROS) production of lung cancer cells. Furthermore, ATAD2 was recently reported as an oncogenic competence factor required for melanoma initiation in melanocytes. Baggiolini et al. [41] have demonstrated that within the melanocytic lineage that starts with undifferentiated neural crest (NC) cells, through the melanoblast (MB) stage, and finally differentiated into melanocytes (MC), only NC cells and MBs have strong transcriptional response following oncogene activation, ultimately resulting in cell transformation into melanoma. In their excellent study, ATAD2 arose as a chromatin remodeling factor required for the establishment of a progenitor signature during cell transformation, ultimately resulting in melanoma formation. Fouret et al. [187] have demonstrated that ATAD2-deregulated expression is a driving force behind Myc’s contribution to uncontrolled cell proliferation in lung adenocarcinoma.

Previously, we reported a consistent positive correlation of ATAD2 expression with cancer stemness across distinct tumor types. We observed that higher-grade tumors displayed significant upregulation of ATAD2. Also, ATAD2-related transcriptome profiles were significantly enriched with known stem cell-derived gene signatures and “cancer hallmark” terms specific for stemness-high tumors, especially the activation of E2F and c-Myc-dependent transcription [118]. This strongly supports the role of ATAD2 in regulating cancer de-differentiation status.

### 4.7. BET Family Transcriptional Co-Activators and Cancer Stemness

BRD4

Among the four mammalian BET family members, BRD4 and BRDT seem more closely related regarding the protein size or level of overall homology, while BRD2 and BRD3 manifest higher similarity to one another than to BRD4 or BRDT [8,26].

BRD4 knockout was previously shown to result in embryonic lethality [188]. Similarly, a lethal phenotype was also observed for BRD2-null animals, although at later stages of embryonic development [189]. In fact, Micco et al. [190] have further demonstrated that only BRD4, and not BRD2 or BRD3, is indisputable for maintaining stem cell self-renewal/pluripotency of ESCs in vitro. BRD4 directly interacts with the OCT-3/4 transcription factor and regulates pluripotency gene expression [191]. Also, BRD4 regulates Nanog expression in ESCs, and Nanog requires BRD4 to maintain the pluripotency of ESCs [192,193].

In cancer, especially regarding cancer stemness, the involvement of BRD4 was most extensively studied. BRD4 is significantly upregulated in a number of cancer types, including melanoma, colon, breast, and bladder cancers [194,195,196]. Recently, Fisher et al. [197] demonstrated that BRD4 drives cancer stem cell-like phenotypes of squamous cell carcinomas by regulating the dNp63a transcription factor, a known facilitator of cell self-renewal. In glioma, BRD4 (and not BRD2 or BRD4) sustained the self-renewal of glioma-initiating cells by modulating the Notch1 signaling pathway, and significant inhibition of BRD4 resulted in the loss of stem cell-like properties of glioma cells [198]. Venkataraman et al. [199] have previously reported that BRD4 inhibition attenuated the self-renewal capacity of tumor cells by suppressing stem cell-associated signaling in Myc-driven medulloblastomas. Another study reported that BRD4 promotes glioma cell stemness via enhancing the activation of WNT/β-catenin signaling (by inducing the miR-142-5p promoter methylation, and for miR-142-5p—Wnt3a as a direct target) [200]. Besides mediating Wnt/β-catenin signaling, BRD4 is also involved in the PI3K-AKT or the Hedgehog (mediated by GLI1) signaling pathways. BRD4 facilitates glioma stem cell (GSC) properties through VEGF/PI3K/AKT signaling, and small molecule inhibitor JQ1 significantly inhibited the self-renewal of GSCs [201]. Later, Wang et al. [202] reported that a combination of BRD4 and HDAC3 inhibitors synergistically halted glioma stem cell growth by inhibiting the GLI1/IL6/STAT3 signaling pathway, suggesting that the repertoire of BRD4-mediated signaling might be more comprehensive.

BRD4 also promotes stemness in cancers of other types, including breast, gastric, esophageal, or prostate tumors. Shi et al. [203] reported a mechanism by which Twist recruits BRD4 to direct WNT5A expression in basal-like breast cancer cells, enhancing their cancer stem cell-like properties. In gastric cancer, BRD4 promotes stemness via attenuating miR-216a-3p-mediated inhibition of the the Wnt/β-catenin signaling pathway [204]. Also, Civenni et al. [205] uncovered a novel link between BRD4, mitochondrial dynamics, and self-renewal of prostate CSCs. Blocking the function of BRD4 robustly impaired the population of CSC in prostate cancer models. Furthermore, in esophageal adenocarcinoma, BRD4 activates the Hippo/YAP1 signaling pathway—one of the primary regulators of cancer aggressiveness and stemness. BRD4 was identified as an essential regulator of YAP1 transcription through direct occupancy of its promoter. Significant inhibition of BRD4 resulted in the depletion of YAP1 and loss of cancer stem cell-like properties [206].

Taken together, BRD4 is a well-documented factor promoting cancer stemness across distinct types of solid tumors, and direct targeting of BRD4 significantly attenuates the cancer stem cell-like properties of tumors.

### 4.8. BrD Proteins with E3 Ubiquitin/SUMO Ligase Activities and Cancer Stemness

TRIM24

TRIM24 (also known as TIF1α) was previously identified as a part of the pluripotency network in mouse embryonic stem cells. Rafiee et al. [207] demonstrated that Trim24 converges with Oct-3/4, Sox2 and Nanog on multiple enhancers and suppresses the expression of developmental genes while activating cell cycle genes, maintaining the mESC population. Also, somatic cells with Trim24 upregulation are more efficiently reprogrammed to iPSCs, which reveals the direct engagement of TRIM24 in establishing pluripotency [208].

TRIM24 has been previously linked to promoting several types of solid tumors, including brain, breast, lung, kidney, ovarian, prostate, esophageal squamous, and head and neck squamous cell carcinomas [209,210,211,212,213,214,215,216,217]. TRIM24 overexpression associates with aggressive malignant phenotypes, suggesting its role in cancer stemness regulation. In fact, Lv et al. [218] demonstrated that TRIM24 is essential to mediate the self-renewal of glioma stem cells in EGFR-driven gliomas. As a transcriptional co-activator of STAT3, TRIM24 leads to the activation of STAT3 downstream signaling in response to EGFR in glioma cells, ultimately supporting the stem cell-like phenotype. Also, TRIM24 promotes stemness and invasiveness of glioblastoma through direct activation of SOX2 expression and the induction of SOX2-targeted transcriptome [219]. In colorectal cancer, TRIM24 overexpression facilitated the in vitro and in vivo growth of CRC tumors, enhanced the stem cell-like characteristics, and upregulated VEGF expression, resulting in enriched angiogenesis. Tian et al. [215] revealed that the activation of the Wnt/β-catenin signaling pathway is necessary for the TRIM24-mediated progression of tumor growth.

TRIM28

TRIM28 is expressed in all cell types, and high TRIM28 expression is observed in embryos and embryonic stem cells. TRIM28 plays a fundamental role in maintaining stem cell pluripotency, at least partially by repressing genes associated with differentiation and inducing the expression of stemness markers [220,221,222]. As previously reported by Klimczak et al. [220], downregulation of TRIM28 expression facilitates the rapid acquisition of a stem-like phenotype upon exogenous expression of Yamanaka’s reprogramming factors, albeit those cells are not sufficient to sustain the stemness.

Moreover, the TRIM28 gene is highly expressed in different cancer types, including breast, glioma, liver, lung, gastric, kidney, ovarian, and pancreatic cancer, and higher expression frequently correlates with poor survival [223,224,225,226,227,228,229,230]. TRIM28 is involved in the transcriptional activation of the EMT program, which mediates the stem cell-like phenotype of breast cancers [231]. Also, together with MAGEA3/6, TRIM28 forms a cancer-specific ubiquitinase that targets AMPK—the “metabolic switch” in cancer, for proteasomal degradation [232]. We have previously demonstrated that the metabolic changes abolished the self-renewal potency of breast cancer stem cells and led to the inhibition of tumor growth upon TRIM28 knockdown [224].

Recently, using the transcriptomic data from distinct types of solid tumors, we have reported that TRIM28 overexpression is significantly associated with an enriched stem cell-like phenotype regardless of the tumor type. Less differentiated tumors characterized by stem cell-like properties exhibit TRIM28 upregulation, and TRIM28-associated transcriptome profiles are robustly enriched with stem cell markers. Surprisingly, this phenomenon was strictly related to TRIM28 and not to other TIF1 family members [118].

TRIM28 is a co-factor for a huge family of KRAB-ZNF transcription factors; some of them were previously recognized in mechanisms governing cancer stemness (reviewed in [233]. Therefore, there are at least two potential modes of action proposed for TRIM28 acting as a facilitator of cancer stemness (in accordance with KRAB-ZNFs): (i) TRIM28 switches off the expression of differentiating genes and/or (ii) TRIM28 enhances the expression of pluripotency markers. However, further studies are needed to clarify the exact role of TRIM28 in cancer stemness across distinct tumor types.

TRIM33

Previously, the TRIM33 protein was shown to regulate the proper differentiation of embryonic stem cells, in contrast to TRIM24 and TRIM28, which sustain self-renewal/pluripotency. Massague et al. [234] demonstrated that the abrogation of TRIM33 expression does not affect stem cell self-renewal, but it impairs the differentiation process. In cancer, TRIM33 was predominantly identified as a tumor suppressor, and low TRIM33 expression correlated with enhanced genomic instability, resulting in cancer progression. Specifically, Pommier et al. [235] have shown that the loss of TRIM33 enhanced tumor aggressiveness by promoting mitotic defects that led to chromosomal abnormalities (increased aneuploidy and chromosome rearrangements). TRIM33 was identified as an effective regulator of TGF-β and Wnt/β-catenin signaling pathways, both of which are indisputable facilitators of stem cell self-renewal [46,236,237]. However, the data clarifying the role of TRIM33 in cancer stemness are still missing.

TRIM66

The last member of the TIF1 family, TRIM66, has recently gained much interest in the context of cancerogenesis. Precisely, TRIM66 was identified as an oncogenic factor in glioma, osteosarcoma, lung, liver, and prostate tumors [238,239,240,241,242]. In normal embryonic stem cells, TRIM66 safeguards their genomic stability [243]. However, in cancer progression, the mode of action of TRIM66 was not linked to its’ chromatin-associated functions. Song et al. [244] revealed that TRIM66 overexpression in glioma plays a vital role in proliferation, apoptosis, and glucose metabolism, possibly by regulating c-Myc/GLUT3 signaling. In osteosarcoma, TRIM66 promoted proliferation and metastasis via the TGF-β signaling pathway and inhibited cell apoptosis by downregulating the TP53 expression in cancer cells [242]. Furthermore, Liu et al. [245] reported that TRIM66 depletion affected the EMT, resulting in an abolished migration and invasive properties of lung cancer cells. The role of TRIM66 in promoting lung cancer progression was later confirmed by Chen et al. [246], who demonstrated that TRIM66 upregulation modulates the level of MMP9 that subsequently induces the TGF-β/SMAD signaling pathway, enhancing tumor aggressiveness. In prostate tumors, TRIM66 promotes malignant progression through the JAK/STAT signaling pathway [238]. In liver tumors, TRIM66 activates the Wnt/β-catenin signaling pathway, and TRIM66 knockdown significantly reduces the proliferation, colony formation, and invasion of HCC cells [241].

Although the abovementioned studies did not directly assess the involvement of TRIM66 in mediating cancer stemness, they strongly imply its stemness-promoting activity. Surprisingly, this finding contrasts with our previously reported work showing a negative association between TRIM66 expression and cancer stemness across most solid TCGA tumors [118]. The transcriptome profiles associated with high TRIM66 expression are robustly depleted with stem cell markers, and this phenomenon is universal regardless of the tumor type. Therefore, further molecular studies are necessary to explain these discrepancies.

### 4.9. ZMYND Transcriptional Co-Repressors and Cancer Stemness

ZMYND8

ZMYND8 was previously recognized as a tumor suppressor in several types of tumors. However, recent studies uncovered its oncogenic potential. Luo et al. [247] have demonstrated that ZMYND8 is selectively expressed in breast cancer stem cells and promotes the EMT, self-renewal of CSCs, and oncogenic transformation through its epigenetic functions. Mechanistically, ZMYND8 is a transcriptional regulator of 27-hydoxycholesterol metabolism, driving breast tumor progression through metabolic reprogramming. Also, Qiu et al. [248] reported that elevated ZMYND8 protein drives the stemness features of bladder cancer, promoting tumor progression, and supporting the ZMYND8 oncogenic function. However, a direct mechanism of ZMYND8-mediated regulation of cancer stemness was not proposed.

In contrast, Mukherjee et al. [249] observed that the loss of ZMYND8 promotes breast cancer stemness, EMT, and drug resistance. ZMYND8 overexpression resulted in the downregulation of tumor-promoting genes by repressing their poised promoters in association with KDM5C and EZH2.

## 5. Conclusions

The role of BrD family members in cancer development and progression has been studied for years, although rarely with relevance to the stem cell compartment of heterogeneous tumors. A number of BrD proteins have garnered scientific interest for many decades, resulting in a relatively prominent role in cancerogenesis, while for other members—the experimental analyses are clearly missing. Here, we summed up the current knowledge about BrD proteins concerning cancer stemness, and also considered its cancer de-differentiation status. Stem cell-associated molecular features of solid tumors are essential in development, progression, therapy resistance, and cancer relapse. Therefore, a novel therapeutic strategy that would directly eradicate this population of cancer cells is urgently needed.

We demonstrated that an increasing number of BrD proteins reinforce cancer stemness, supporting the maintenance of the cancer stem cell population in vitro and in vivo. Next to several well-known cancer stemness facilitators, including TRIM28 or BRD4, various other BrD proteins were recently proven to sustain stem cell-like phenotype via the utilization of distinct mechanisms. However, for most of the BrD members, further work is indisputable to fully characterize their contribution to cancer stem cell self-renewal machinery.

In regard to the significant druggability of bromodomain, it would be of the highest importance to determine whether specific BrD protein members could serve as therapeutic targets in cancers exhibiting de-differentiated tumor characteristics. Ultimately, inhibition of those BrDs might result in the disruption of stemness-mediating machinery and, consequently, abolish tumor growth and prevent tumor relapse.

## Figures and Tables

**Figure 1 ijms-24-00995-f001:**
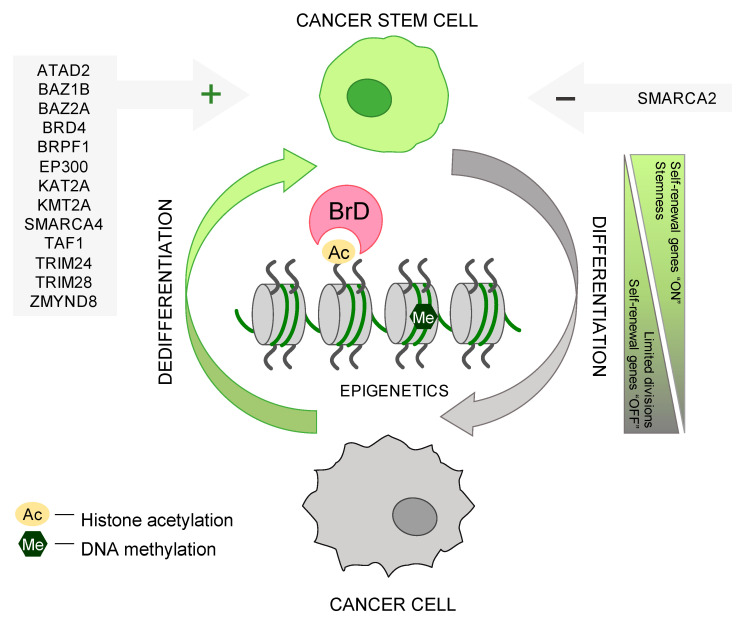
Bromodomain (BrD) proteins and their engagement in regulating the cancer stem cell-like phenotype of solid tumors. BrD module recognizes ε-N-lysine acetylation (Kac) motifs on histone proteins, enabling BrD proteins to mediate epigenetic regulation of gene expression. To date, several BrD members were confirmed mechanistically as positive regulators of cancer stemness, including ATAD2, BAZ1B, BAZ2A, BRD4, BRPF1, EP300, KAT2A, KMT2A, SMARCA4, TAF1, TRIM24, TRIM28, and ZMYND8. On the other hand, SMARCA2 was recognized as a negative regulator of cancer stemness.

## Data Availability

Not applicable.

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
