# Peer review of "Bromodomain (BrD) Family Members as Regulators of Cancer Stemness—A Comprehensive Review"

_ijms, 2023, doi:10.3390/ijms24020995_

Round 1

Reviewer 1 Report

The present manuscript entitled 'Bromodomain (BrD) family members as regulators of cancer stemness" by Patrycja Czerwinska and Andrzej Adam Mackiewicz is very informative and timely.  Although, BrD family members in cancer development and progression has been studied for years, authors demonstrated that an increasing number of BrD proteins reinforce cancer stemness, supporting the maintenance of the cancer stem cell population both in vitro and in vivo. Additionally, it highlights the importance of BrD proteins as a potential novel therapeutic targets in cancers.

 I have no further comments to improve the quality of the manuscript.

Author Response

We appreciate the time and effort the Reviewer has dedicated to revising our manuscript. We are grateful for the positive feedback.

Reviewer 2 Report

The manuscript  "Bromodomain (BrD) family members as regulators of cancer stemness – a comprehensive review " is an important review as although a large amount of  papers related with the subject , a consolidated review including all BrD members discussing the informations already clear and those that need more studies about each of them was missing.

Author Response

We appreciate the time and effort the Reviewer has dedicated to revising our manuscript. In this comprehensive review, we demonstrated that an increasing number of Bromodomain-encoding (BrD) proteins reinforce cancer stemness, supporting the maintenance of the cancer stem cell (CSC) population in vitro and in vivo. Specifically, we pointed at not-so-commonly recognized BrD members and their involvement in sustaining stem cell-like phenotype (next to several well-known cancer stemness facilitators, including TRIM28 or BRD4).In this review, we highlight the value of the BrD protein family in supporting cancer dedifferentiation and imply an urgent need for developing BrD-specific inhibitors to eradicate the CSC-like population directly.

Reviewer 3 Report

Dear authors,

I would like to congratulate with you for this original and well-made review. This paper is in accordance with journal topic, conceptually solid and contemporary.

You have been addressing your aim with the descriptions of the current epigenetic mechanisms of cancer regulation and maintenance.

In my opinion this manuscript is ideally valid and can be accepted after a few tweaks.

Minor points:

-        Please, integrate N. 1 figure acting as grafical abstract, and describe your aim in an illustrative manner.

-        As in paragraph 2 (Cancer stemness “in vitro” – Line 62) it would be appropriate to add the state of art on the aspects of Cancer stemness in vivo, to give a broader view to the reader.

Moreover, it might be interesting for the reader to get an updated overview of the other examples of proteins that modify the structure of DNA, modulating the cell phenotype. Please see and eventually cite: DOI 10.3389/fendo.2022.1051988

Best,

Author Response

We appreciate the time and effort the Reviewer has dedicated to revising our manuscript. We are happy about the positive feedback. 

Below are the replies to the Reviewer's suggestions:

Reviewer: Please, integrate N. 1 figure acting as grafical abstract, and describe your aim in an illustrative manner.

Reply: According to the Reviewer’s suggestions, Figure 1 was added to Section 4. Several BrD family members play a fundamental role in cancer stem cell maintenance.

Reviewer: As in paragraph 2 (Cancer stemness “in vitro” – Line 62) it would be appropriate to add the state of art on the aspects of Cancer stemness in vivo, to give a broader view to the reader.

Reply: Paragraph 2 was modified to summarize the in vivo assays used for cancer stemness evaluation. All modifications to the text are highlighted in yellow.

Reviewer: Moreover, it might be interesting for the reader to get an updated overview of the other examples of proteins that modify the structure of DNA, modulating the cell phenotype. Please see and eventually cite: DOI 10.3389/fendo.2022.1051988

Reply: In our paper, we specifically focus on the family of bromodomain-encoding proteins and their involvement (as epigenetic regulators) in maintaining cancer stemness. It is interesting to analyze the role of other factors that mediate epigenetic regulation of gene expression in regulating the cancer stem cell population. This is our current venture that will be published as a separate paper.